# Mitigation of PFOA-Induced Developmental Toxicity in *Danio rerio* by *Bacillus subtilis* var. *natto*: Focus on Growth and Ossification

**DOI:** 10.3390/ijms26094261

**Published:** 2025-04-30

**Authors:** Christian Giommi, Marta Lombó, Francesca Francioni, Fiorenza Sella, Hamid R. Habibi, Francesca Maradonna, Oliana Carnevali

**Affiliations:** 1Department of Life and Environmental Sciences, Polytechnic University of Marche, 60131 Ancona, Italy; c.giommi@staff.univpm.it (C.G.); mloma@unileon.es (M.L.); f.sella@pm.univpm.it (F.S.); 2INBB—Biostructures and Biosystems National Institute, 00136 Roma, Italy; 3Department of Molecular Biology, Faculty of Biology and Environmental Sciences, University of León, 24071 León, Spain; 4Department of Agricultural, Food and Environmental Sciences, Marche Polytechnic University, 60131 Ancona, Italy; f.francioni@pm.univpm.it; 5Department of Biological Sciences, University of Calgary, Calgary, AB T2N 1N4, Canada; habibi@ucalgary.ca

**Keywords:** per- and polyfluoroalkyl substances, probiotics, zebrafish, skeletal malformations, craniocephalic alterations

## Abstract

Perfluorooctanoic acid (PFOA) is a persistent environmental contaminant that resists biological degradation and accumulates in organisms. It disrupts zebrafish embryo development, affecting their heartbeat rate and locomotion. Meanwhile, probiotics are known to enhance the development and ossification of zebrafish embryos. In this study, we examined the toxic effects of PFOA on growth and bone formation in zebrafish and the potential of the probiotic *Bacillus subtilis* var. *natto* to counteract its toxicity. Larvae were exposed to 0, 50, or 100 mg/L PFOA from hatching to 21 days post-fertilization (dpf), with or without dietary probiotic supplementation (10^7^ CFU/larva/day), and they were sampled at 7, 14, and 21 dpf. PFOA exposure reduced standard length at 21 dpf, while the co-administration of probiotics mitigated these effects. Craniofacial cartilage defects appeared in larvae exposed to 50 mg/L PFOA at 7 and 14 dpf, while 100 mg/L PFOA impaired bone development at 7 dpf. Probiotics counteracted these abnormalities. PFOA also delayed ossification, correlating with the downregulation of *col10a1a*, *runx2b*, and *cyp26b1*, while the probiotic treatment restored normal ossification. These findings improve our understanding of PFOA’s detrimental effects on zebrafish growth and bone formation while demonstrating the protective role of probiotics against PFOA-induced developmental toxicity.

## 1. Introduction

Per- and poly-fluorinated substances (PFASs) are industrial chemicals widely used in everyday items, including nonstick cookware coatings, waterproof and stain-resistant materials, food packages, and firefighting foams. Among PFAS compounds, perfluorooctanoic acid (PFOA) was one of the most extensively produced until it was banned globally in 2019 by the Stockholm Convention on Persistent Organic Pollutants [1] and in 2020 in Europe by REACH regulations [2]. However, these restrictions did not stop the use of PFOA in certain medical devices, firefighting foams, and protective clothing coating. As a result, PFOA continues to pose environmental and public concerns due to its high persistence, resistance to degradation, and bioaccumulation capacity. It easily leaches from household items, leading to its widespread presence in air, water, soil, and biota. Notably, PFOA has been found to negatively impact growth [3], early development [4,5] and ossification [6] in zebrafish, raising significant concerns about its exposure, particularly in early life stages. In fact, PFOA gavage for 28 days at 1 mg/kg/day or 2 mg/kg/day has been associated with bone growth retardation and skeletal malformations in male BALB/c mice [7]. One of the key mechanisms through which PFOA exerts these effects is its endocrine-disrupting activity, particularly through interference with thyroid function, which regulates different developmental processes including ossification [8,9]. This endocrine disruptor activity was evidenced by in vitro studies on rat thyroid cells (FRTL-5) exposed to PFOA at concentrations of 0, 1.25, 5, 20, and 80 μM for 24 h and 48 h [10] or on FRTL-5 and primary normal human thyroid (NHT) cells exposed to 0.1, 1, 10, 100 and 1000 ng/mL PFOA for 24, 48, and 72 h [11].

Zebrafish represent a valuable experimental model for investigating the toxic effects of pollutants on development [12,13,14,15,16]. In zebrafish, intramembranous ossification (which is involved in skull bone ossification), perichondral ossification (which is involved in jaw ossification), and endochondral ossification (which is involved in the formation of vertebrae and fin bones) are concomitantly present [17]. Perichondral and endochondral ossification begins with the production of collagen by hypertrophic chondrocytes, followed by extracellular matrix mineralization carried out by osteoblasts. During this process, *col10a1a*, which encodes Collagen Type X Alpha 1a, plays a crucial role in collagen deposition by hypertrophic chondrocytes during osteogenesis [18]. In fact, *col10a1a^−/−^* mutant zebrafish larvae showed reduced chondrocranium size and impaired bone mineralization, effects that persisted into adulthood, leading to decreased vertebral thickness and tissue mineral density [19]. Another key gene, *runx2b*, which encodes Runt-related transcription factor 2, works alongside *col10a1a* to regulate cartilage-to-bone transition. *Runx2b* is expressed by hypertrophic chondrocytes to promote the differentiation of osteoblasts from mesenchymal cells, thereby facilitating extracellular matrix mineralization [20,21]. Additionally, *cyp26b1*, which encodes cytochrome P450 26B1, an enzyme responsible for retinoic acid (RA) metabolism, plays a critical role in skeletal development [22,23]. Cyp26b1 degrades RA, preventing its excessive accumulation, which could otherwise disrupt ossification. This gene is essential for proper craniofacial bone ossification, as it affects bones derived from neural crest cells [24,25] and is also involved in the ossification of the vertebral column and fin endoskeleton [25,26]. Overexpression of *cyp26b1* can delay ossification due to excessive RA degradation, hindering chondrocyte maturation and osteoblast differentiation [23]. Conversely, *cyp26b1* loss-of-function mutants display skeletal malformations due to premature cartilage resorption, defective vertebral ossification, and abnormal craniofacial development [27].

While the detrimental effects of PFOA and its substitutes on zebrafish development [28,29,30,31,32] have been reported, probiotic administration has been shown to enhance growth and ossification [33,34,35]. These effects have been thoroughly investigated in vitro [36,37] and using animal models [38,39,40,41] including *Danio rerio* [42,43,44]. Notably, *Bacillus subtilis* var. *natto* has been shown to enhance zebrafish larval growth and ossification [45] and to promote bone regeneration following injury in adult zebrafish specimens [26]. Moreover, probiotic administration has been reported to mitigate the adverse effects of endocrine-disrupting chemicals at different biological levels [46,47,48,49,50,51,52]. Specifically, another probiotic strain, *Lactobacillus rhamnosus*, has been shown to counteract the negative effects of another PFAS, perfluorobutanesulfonate (PFBS), on zebrafish larval growth [53,54,55]. On this basis, replacing PFOA with other chemicals is not an effective approach for reducing the environmental impact of PFAS. The strategies adopted so far by certain manufacturers have involved devising substitute compounds that have similar or even greater levels of toxicity. Therefore, probiotic administration could serve as a promising strategy to mitigate PFAS-induced toxicity.

In this context, the present study hypothesizes that the administration of the probiotic *Bacillus subtilis* var. *natto* could mitigate any potential negative effects induced by PFOA exposure on zebrafish development specifically, on bone formation and mineralization. To test this hypothesis, integrated morphological and molecular studies were conducted over different crucial developmental stages, encompassing different time points of ossification: early ossification, with most skeletal elements being cartilaginous (7 dpf), the onset of mineralization when jaw bones and vertebrae start to show mineralization (14 dpf), and advanced ossification, with most skeletal elements showing mineralization (21 dpf).

## 2. Results

### 2.1. Impact of PFOA Exposure and Probiotic Treatment on Zebrafish Growth

Regarding larval growth, the most interesting results were observed at 21 dpf: the weight of the larvae was increased following treatment with the probiotic, either alone or with PFOA (Figure 1a); however, the standard length of the larvae only increased when the probiotic was administered alone (Figure 1b). Exposure to both concentrations of PFOA significantly reduced the standard length at 21 dpf (Figure 1b). The observed PFOA-induced response was completely reversed when the larvae were co-treated with *B. subtilis* var. *natto* (Figure 1b).

Focusing on genes involved in zebrafish growth and development, myostatin a (*mstna*), thyroid hormone receptor alpha a (*thraa*), and the insulin growth factors *igf1* and *igf2* were analyzed at each time point (Figure 1c and Appendix A). The results demonstrate that exposure to both concentrations of PFOA reduced the *thraa* transcript at 7 dpf. This alteration was partially counteracted by the probiotic treatment (Figure 1c). The mRNA levels of the other genes investigated were not affected following treatments at 7 (Figure 1c), 14 or 21 dpf (Appendix A).

### 2.2. Effects of PFOA Exposure and Probiotic Treatments on Zebrafish Larval Morphology

The ocular morphometric assessment revealed that, at 14 dpf, the larvae exposed to 100 mg/L PFOA and treated with probiotic (100 PFOA + P) showed a significant increase in eye angle (Figure 2a) and eye distance (Figure 2b) compared to all other groups. At 21 dpf, both PFOA concentrations induced increases in eye diameter that were counteracted by concomitant treatment with the probiotic (Figure 2c). The eye depth did not change among the groups (Figure 2d).

To further investigate the capacity of PFOA to induce craniocephalic malformations, the length of head cartilages (Appendix A) and the angles between these cartilages (Figure 3) were analyzed at all time points tested using alcian blue and alizarin red staining. Although the length of head cartilages was not affected by treatments (Appendix A), exposure to both PFOA concentrations increased the Meckel’s-Meckel angle at 7 dpf, an effect mitigated by the co-administration of the probiotic. At 14 dpf, the Meckel’s-Meckel’s angle was only increased following exposure to 50 mg/L PFOA (50 PFOA), and this alteration was also counteracted by probiotic co-administration (Figure 3c). In 21-dpf larvae, an increase in the Meckel’s-Meckel’s angle and a decrease in the PQ–Meckel angle were observed following exposure to 50 mg/L PFOA concomitantly with probiotic administration (50 PFOA + P) and 100 PFOA + P groups (Figure 3c,d). Moreover, otolith and cleithrum formation was observed in the control group (C), the *Bacillus subtilis* var. *natto* administered group (P), and the 100 mg/L PFOA (100 PFOA) and 100 PFOA + P groups at 7 dpf (Figure 3e), although the ossification of these structures was absent in the 50 PFOA group, either alone or with a probiotic treatment.

### 2.3. Effects of PFOA Exposure and Probiotic Treatment on Zebrafish Larval Ossification

To further investigate the adverse effects of PFOA on ossification, the mRNA levels of genes involved in ossification, including Collagen type X alpha 1a (*col10a1a*), RUNX family transcription factor b (*runx2b*), Cytochrome P450, family26 subfamily b polypeptide 1 (*cyp26b1*), and secreted phosphoprotein 1 (*spp1*), were analyzed at each time point (Figure 4a,b and Appendix A). Exposure to 50 PFOA increased the *cyp26b1* mRNA level at 14 dpf compared to all other groups (Figure 4a) and decreased the *cyp26b1* transcript level at 21 dpf compared to C (Figure 4b). In addition, 50 PFOA caused a decrease in both *col10a1a* and *runx2b* transcript levels compared to C (Figure 4b). After the co-administration of 50 PFOA and P, *col10a1a* expression remained decreased compared to C. In contrast, the *runx2b* transcript level increased compared to all the other groups, following an opposite trend to that of 50 PFOA alone (Figure 4b). Furthermore, the alizarin red/alcian blue staining conducted at 21 dpf showed similar ossification of the pterygiophore vertebral arches, hypurals, and urals in both the C and P groups. However, exposure to both concentrations of PFOA reduced ossification of these bones. Notably, the P completely rescued this phenotype when co-administered with 100 PFOA, while only partially rescuing it at the lowest concentrations (Figure 4c,d).

## 3. Discussion

Using a morphological and molecular approach, we provided clear evidence that the administration of a probiotic can counteract PFOA-induced skeletal abnormalities. The results obtained align well with previous observations regarding the ability of PFOA [3,4,56,57], as well as other PFASs [6,56,57,58,59], to affect growth and development, as well as heart and swim bladder defects [6]. The present study confirmed that exposure to PFOA alters larval growth by decreasing the standard length. Administration of probiotics for 21 days, however, resulted in an overall increase in the larval body growth, in agreement with a vast body of literature highlighting the ability of the probiotic *Bacillus subtilis* var. *natto* [45], as well as other probiotics such as *L. rhamnosus* [42,43,60,61,62], to enhance fish growth. The results were also supported by the increase in the wet weight of the larvae, which was also evident when this probiotic was co-administered with PFOA exposure. It should be noted that the standard-length alteration induced by PFOA was mitigated by the co-administration of a probiotic. To date, no data are reported in the literature concerning the mitigation effects of probiotics against PFOA-induced growth alterations. However, previous studies in zebrafish have demonstrated the capacity of *L. rhamnosus* to counteract developmental alterations induced by PFBS, an alternative PFAS, at concentrations of 1, 3.3, and 10 mg/L [53,54] and 10 µg/L [55], by restoring the larval length, weight, and growth rate to those of unexposed fish. To investigate the possible molecular mechanisms underlying the changes herein observed on larvae biometry, we conducted a deeper analysis focusing on key genes involved in growth. While alterations in growth were mainly observed at 21 dpf, these changes were not associated with a modulation of the transcript levels of the genes investigated. This is not surprising, as phenotypic changes may not always happen at the time of changes in gene expression. Nevertheless, considering the pivotal role of thyroid hormones in the regulation of growth [63,64,65,66] and the well-documented disruptive effects of PFAS, including PFOA, on the thyroid axis, and in turn on growth and development [4,56,58,59,67], we analyzed the mRNA levels of *thraa* at all of the selected time points. The *thraa* transcript level was reduced at 7 dpf in all the groups exposed to PFOA. Thus, we postulate that the alteration in growth observed may in part be caused by dysregulation of the thyroid pathway during early development. In this context, the thyroid system is critical to eye development and function [68,69,70], and alterations in the thyroid system could, to a certain extent, be one reason for the observed alterations in eye diameter. Indeed, eye morphology alterations were observed upon exposure to several EDCs [71]. Thyroid-hormone-disrupting chemicals such as propylthiouracil (PTU) and tetrabromobisphenol A (TBBPA) were reported to alter eye size and pigmentation and to induce changes in the retinal cellular structure in zebrafish embryos [72]. Similarly, exposure to bisphenol S (BPS) [73] altered retinoid metabolism and thyroid hormone homeostasis, and bisphenol A (BPA) [74] disrupted retinal layering in zebrafish larvae. In addition, triclosan and benzophenone-2 exposure led to thyroid follicle hyperplasia, leading to an alteration in the retina’s cellular structure in zebrafish larvae [75]. The increase in eye diameter observed in this study is consistent with similar changes reported in zebrafish exposed to a plasticizer, BPA, during development [76]. The mitigation of eye diameter exerted by the co-administration of *Bacillus subtilis* in PFOA-exposed larve adds new evidence to the well-established beneficial role of different probiotic strains, as well as *Bacillus subtilis* var. *natto*, in promoting developmental processes and in counteracting the detrimental effects of endocrine disruptors on larval development.

A similar alteration in jaw morphology to that observed in the present study following exposure to PFOA has previously been described in zebrafish exposed to BPA, increasing the Meckel–Meckel angle during early development until 120 h post fertilization [76]. The mitigation of PFOA toxicity following the administration of *Bacillus subtilis* var. *natto* highlights the ability to promote bone ossification, as previously observed in caudal fin regeneration [26] and in backbone ossification [34]. To deepen our knowledge regarding the possible mechanism of probiotic action, we evaluated the transcript levels of key genes involved in ossification. While the co-administration of probiotics partially mitigated the effects of PFOA on *col10a1a*, it led to an increased transcript level of *runx2b* at 21 dpf. Since *runx2b* directly regulates bone formation in the jaws via a cartilage precursor during perichondral ossification [20,21], this increase may underline the combined effects of PFOA and the probiotic on the Meckel’s-Meckel’s and PQ–Meckel’s angles observed at 21 dpf. This interaction could have contributed to the persistence of alterations induced earlier in development. Notably, in the present study, the lowest concentration of PFOA tested reduced both *col10a1a* and *runx2b* transcript levels, correlating with the reduced ossification observed via alcian blue and alizarin red staining at 21 dpf. These data suggest a non-monotonic response of PFOA, which has already been reported for other EDCs, such as BPA and phthalates [77,78]. Along with *col10a1a* and *runx2b*, *cyp26b1*, an additional master gene involved in chondrocyte maturation, osteoblast differentiation, and cartilage-to-bone transition, was deregulated. Given the role of this gene, the craniofacial malformations and ossification delay induced by the lowest PFOA concentration could be attributed to the increased *cyp26b1* transcript levels found in this group at 14 dpf. This likely led to higher CYP26b1 and thus to a major RA catabolism, reducing its availability for ossification processes [27]. Furthermore, the decrease in *cyp26b1* in the same group at 21 dpf could be responsible for prolonged cartilage maintenance and delayed ossification [23], as evidenced by the alizarin red/alcian blue staining histological analysis. Notably, probiotic administration was able to mitigate this effect, as was particularly evident at 14 dpf and partially at 21 dpf, when *col10a1a*, *cyp26b1*, and *runx2b* transcript levels became closer to those of the control group, further indicating the capacity of probiotic administration to mitigate PFOA-induced toxicity on ossification.

Preventing the toxic effects of PFOA on ossification is essential to reducing the risk of skeletal abnormalities and long-term health issues, particularly in developing individuals. Overall, this study not only advances knowledge about PFOA toxicity on ossification but also provides a promising foundation for developing intervention strategies to reduce pollutant-related developmental defects. This highlights the potential of probiotics as a natural strategy to counteract PFOA-induced toxicity and support proper bone development, offering a promising approach to mitigating the adverse effects of environmental contaminants on the skeletal system.

## 4. Materials and Methods

### 4.1. Zebrafish Husbandry and Treatment

Adult zebrafish (AB strain) were kept under controlled conditions (28 ± 1 °C with a 14:10 light:dark period) in a recirculation system and fed twice per day with an adult zebrafish complete diet (Zeigler Bross, Inc., Gardners, PA, USA, Aquamerik Inc., St. Nicolas, QC, Canada) and live brine shrimp nauplii, Artemia cysts. In total, 3000 embryos were obtained from synchronized breeding; they were selected and maintained under standard conditions until hatching (3 dpf) in fish water from a reverse osmosis system. At this point, the embryos were equally divided into 6 experimental groups in triplicate: (1) control (C), untreated; (2) probiotic *Bacillus subtilis* var. *natto* administration via water at a final concentration of 10^7^ CFU/mL (P); (3) exposed to 50 mg/L PFOA (50 PFOA); (4) exposed to 50 mg/L PFOA plus the administration of *Bacillus subtilis* var. *natto* via water at a final concentration of 10^7^ CFU/mL (50 PFOA + P); (5) exposed to 100 mg/L PFOA (100 PFOA); (6) exposed to 100 mg/L PFOA plus the administration of *Bacillus subtilis* var. *natto* via water at a final concentration of 10^7^ CFU/mL (100 PFOA + P). Rearing water was completely changed daily, and PFOA and the probiotic were renewed during every water change, being handled identically across groups to minimize co-morbid conditions. The experiment lasted from hatching till 21 dpf. The PFOA concentrations were selected considering previous studies’ reports of its ability to alter zebrafish development [4] and zebrafish female fertility [79], to induce peroxisomal fatty acid oxidation, and to alter the immune system in *Oryzias latipes* adult liver [80], and considering the ability of this pollutant to alter the development and reproduction of both the amphipod *Hyalella azteca* and the fish *Pimephales promelas* [81]. Regarding the probiotic, the concentration was selected based on previous studies in zebrafish using other probiotic strains such as *Lactobacillus acidophilus* AC [82] and mice with *Lactobacillus rhamnosus GG* [83], *Lactobacillus rhamnosus* MTCC-5897 [84], *Weissella cibaria* [85], and *Bifidobacterium breve* [86]. Larvae were fed twice a day with rotifers starting from day 5 days post fertilization (dpf) until the end of the experiment, and the water was changed and renewed in all tanks every day. To follow the effects throughout the development, three sampling timings were established at 7, 14, and 21 dpf (Figure 5).

All procedures involving animals were conducted according to the University of Calgary animal care protocols (AC19-0160 approved on 8 March 2022) for the care and use of experimental animals. All efforts were made to minimize animal suffering. Larvae were euthanized in a solution containing 300 mg/L MS-222 (3-aminobenzoic acid ethyl ester; Sigma Aldrich, St. Louis, MO, USA) buffered to pH 7.4, according to the University of Calgary animal care protocol. For each time point, 30 larvae were fixed with 4% paraformaldehyde (PFA) (Bio-Optica, Milan, Italy) for 2 h, washed three times with 70% ethanol, and then stored at 4 °C until being processed for biometric evaluations and alcian blue and alizarin red staining. Additionally, five pools of 25 larvae each were sampled and stored at −80 °C until being processed for total RNA extraction.

### 4.2. Biometric Evaluation

Larvae length and wet weight were measured following the protocol of Zarantoniello et al. [87] to assess the presence of biometric changes among the experimental groups. The length of five larvae per group at each time point was measured using a caliber (Measy 2000 Typ 5921, Greifensee, Switzerland; precision: 0.1 mm). For the wet weight measure, three replicates of five larvae per group at each time point were used employing an OHAUS Explorer (OHAUS Europe GmbH, Greifensee, Switzerland) analytical balance (precision: 0.1 mg).

### 4.3. Alcian Blue and Alizarin Red Staining

To analyze both the cartilage and the bone, alcian blue/alizarin red staining was performed according to the protocol described by Walker and Kimmel [88] and as previously described [89] on 15 larvae per each group and time point.

### 4.4. Morphological Studies

Larval malformations, as well as ocular and jaw morphometrical measurements, were studied via stereomicroscopy (Leica microsystems, Wetzlar, Germany). The craniocephalic morphological measurements (ceratohyal (CH) length, palatoquadrate (PQ) length, CH–Meckel length, CH–CH angle, CH–PQ angle, PQ–Meckel angle and Meckel-Meckel angle) were conducted on larvae stained with alcian blue/alizarin red using ImageJ-win64 (Image J, version 1.54, NIH, Bethesda, MD, USA, https://imagej.nih.gov/ij/, accessed on 6 November 2023). An optical microscope (Zeiss imager M.2, Castiglione Orona, Italy) coupled with a photo camera (Axiocam 105) was used to obtain the representative images of each experimental group at the different time points.

### 4.5. Gene Expression Analysis

Total RNA was extracted using RNAzol^®^ RT (Qiagen Science, Milano, Italy) following the manufacturer’s instructions from 7, 14, 21 dpf larvae from each experimental group using 5 pools of 25 larvae per group at each time point. Final RNA concentrations were determined using the Nanophotometer (Implen GmbH, Munich, Germany), and RNA integrity was verified on 1% agarose gel (Appendix A). To avoid genomic DNA contamination, an additional step to selectively digest DNA was performed using a DNAse kit (Sigma-Aldrich, St. Louis, MO, USA). Then, cDNA synthesis was performed using an iScript cDNA Synthesis Kit (Bio-Rad Laboratories, Milano, Italy) according to the manufacturer’s protocol. Real-time PCR was performed in a thermal cycler (CFX Connect, Bio-Rad Laboratories, Milano, Italy) in duplicates. Forward and reverse primers were diluted at a final concentration of 10 pmol/mL. Ribosomal protein, large, P0 (*rplp0*) and ribosomal protein L13a (*rpl13a)* were used as endogenous reference genes. Primer sequences, accession numbers, and annealing temperatures are reported in Table 1. The data were analyzed using Bio-Rad CFX maestro software 2.2 (5.2.008.0222) RTq-PCR data were analyzed using the Gene Expression Analysis for iCycler iQ^®^ Real-Time PCR Detection System provided by BioRad (Milan, Italy).

### 4.6. Statistical Analysis

All data were analyzed using a two-way ANOVA with treatment and developmental stage as factors, followed by Tukey’s multiple comparisons test (GraphPad Prism 8.0, San Diego, CA, USA). Post hoc comparisons were performed within each developmental stage to evaluate differences among treatments. Statistical significance was set at *p* < 0.05.

## 5. Conclusions

The present study demonstrates the toxicity of PFOA on the development of zebrafish and its harmful impact on bone formation through delayed bone ossification and skeletal malformations. Remarkably, three master genes involved in ossification were affected by the lowest concentration of PFOA, which emerged as the most detrimental, highlighting the non-monotonic response of this toxicant. The probiotic *Bacillus subtilis* var. *natto* counteracted PFOA-induced growth impairments and improved ossification, aligning with previous evidence on the benefits of probiotics in fish development. To the best of our knowledge, this is the first evidence of probiotic mitigation of the adverse impact of PFOA on development and ossification. Future research is needed to explore the potential mitigation capacity of different probiotic strains and formulations. Additionally, in vitro investigations using 3D models that mimic the bone microenvironment would provide valuable insight into the deleterious effects of PFAS on osteogenesis and their association with bone-related diseases.

## Figures and Tables

**Figure 1 ijms-26-04261-f001:**
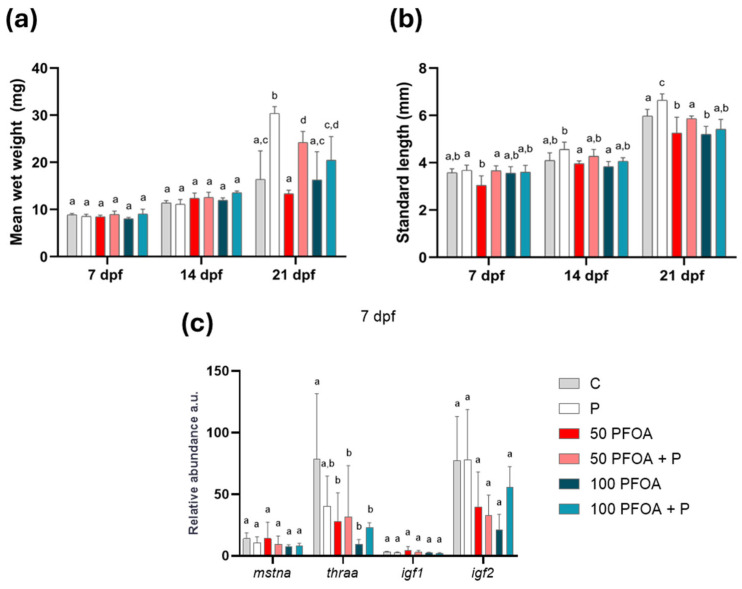
Zebrafish growth assessment. (**a**) Mean wet body weight and (**b**) standard length in the different experimental groups at 7, 14 and 21 dpf. Data are reported as means ± SD. Different letters indicate statistically significant changes (*p* < 0.05) among the experimental groups at each time point (*n* = 3 for mean wet weight and 5 for standard length). (**c**) Histogram showing the *mstna*, *thraa*, *igf1*, and *igf2* transcript levels in the different experimental groups at 7 dpf reported as relative abundance in arbitrary units (a.u). Data are reported as means ± SD. Different letters indicate statistically significant changes (*p* < 0.05) among the experimental groups (*n* = 5).

**Figure 2 ijms-26-04261-f002:**
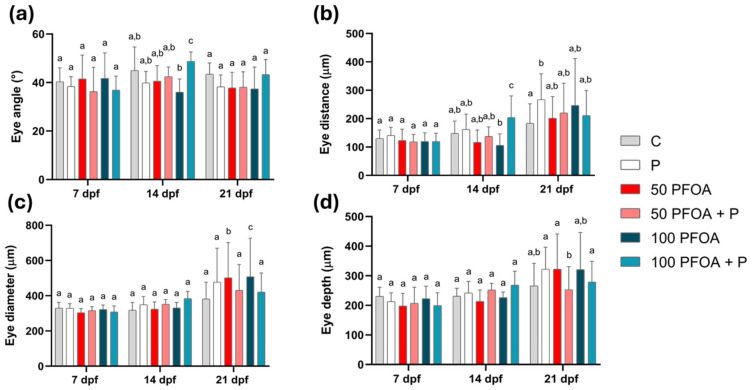
Zebrafish ocular morphometric analysis. (**a**) Eye angle (degrees), (**b**) eye distance (µm), (**c**) eye diameter (µm), and (**d**) eye depth (µm) in the different experimental groups at 7, 14 and 21 dpf. Data are reported as means ± SD. Different letters indicate statistically significant changes (*p* < 0.05) among the experimental groups (*n* = 15) at each time point.

**Figure 3 ijms-26-04261-f003:**
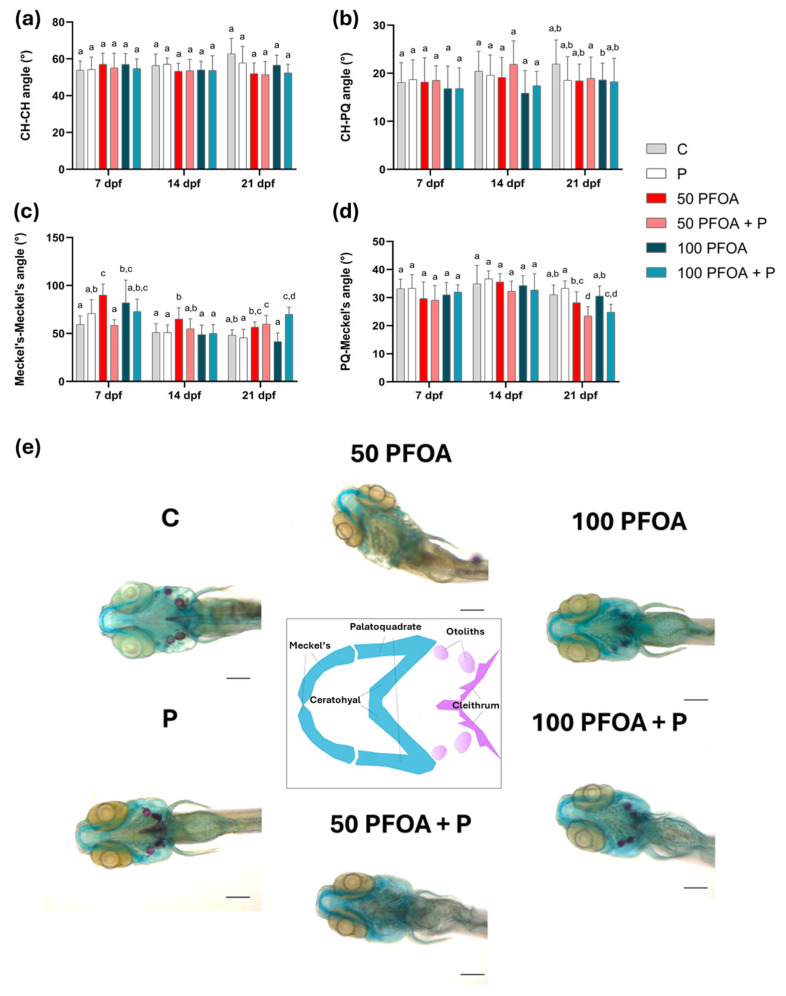
Zebrafish craniocephalic morphometric analysis. (**a**) CH–CH angles (degrees), (**b**) CH–PQ angles (degrees), (**c**) PQ–Meckel angle (degrees) and (**d**) Meckel’s-Meckel’s angle (degrees) in the different experimental groups at 7, 14, and 21 dpf. Data are reported as means ± SD. Different letters indicate statistically significant changes (*p* < 0.05) among the experimental groups (*n* = 15) at each time point. (**e**) Representative microphotographs of Alcian-blue- and Alizarin-red-stained head at 7 dpf. Scale bar = 200 µm.

**Figure 4 ijms-26-04261-f004:**
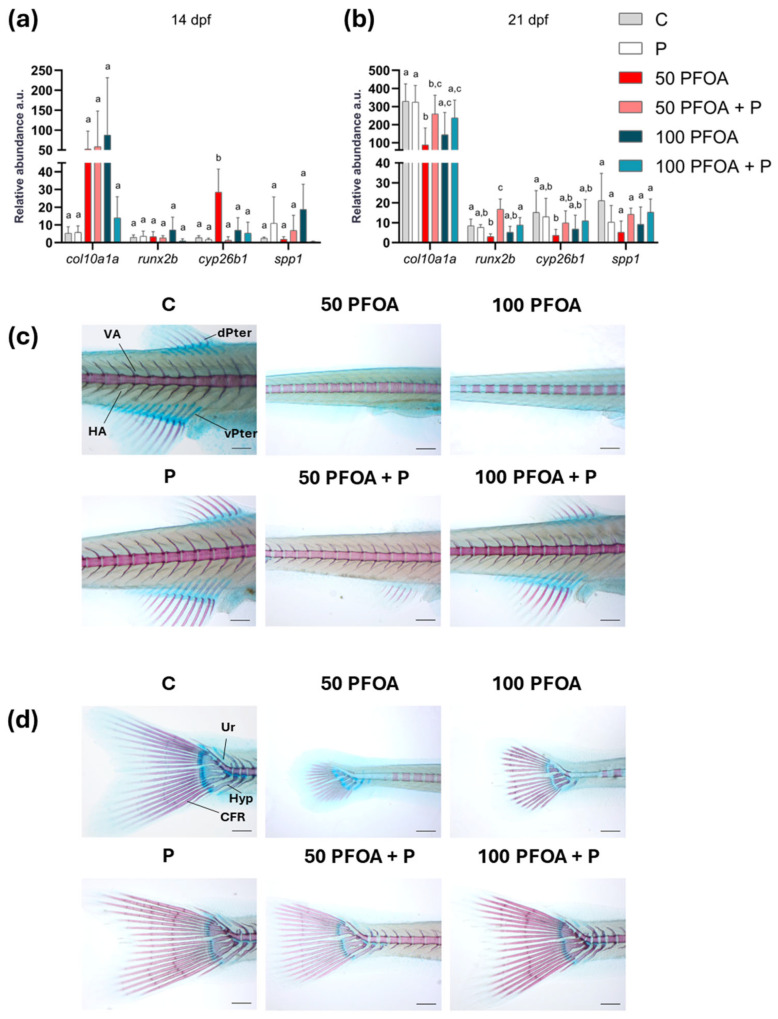
Zebrafish ossification assessment. Histograms summarizing the *col10a1a*, *runx2b*, *cyp26b1*, and *spp1* transcript levels in the different experimental groups at (**a**) 14 and (**b**) 21 dpf as relative abundance in arbitrary units (a.u). Data are reported as means ± SD. Different letters indicate statistically significant changes (*p* < 0.05) among the experimental groups (*n* = 5). Representative images or (**c**) pterygiophore and arch ossification and (**d**) hypural and ural ossification in the different experimental groups at 21 dpf by Alcian blue (cartilages stained in blue) and Alizarin red (bones stained in red) staining. Scale bar = 200 µm. Ur = ural; Hyp = hypural; CFR = caudal fin ray.

**Figure 5 ijms-26-04261-f005:**
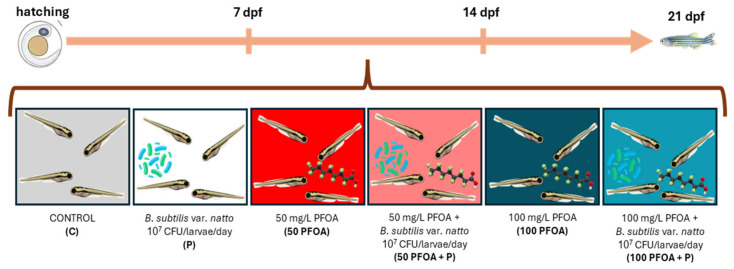
Experimental design. At hatching, larvae were divided into 6 experimental groups: C, P, 50 PFOA, 50 PFOA + P, 100 PFOA, 100 PFOA + P. To trace the detrimental and/or beneficial effects during development, three sampling timings were established at 7, 14, and 21 dpf.

**Table 1 ijms-26-04261-t001:** List of primers designed for the gene expression analysis via RT-qPCR.

Gene Name	Gene Symbol	Primer Sequence (5′-3′)	Accession Number	Tm (C°)
Ribosomal protein large P0	*rplp0*	F: CTGAACATCTCGCCCTTCTC R: TAGCCGATCTGCAGACACAC	NM_131580.2	60
Ribosomal protein L13a	*rpl13a*	F: TCTGGAGGACTGTAAGAGGTATGC R: AGACGCACAATCTTGAGAGCAG	NM_212784.1	59
Insulin-like growth factor 1	*igf1*	F: GGCAAATCTCCACGATCTCTAC R: GGCAAATCTCCACGATCTCTA	NM_131825.2	53
Insulin-like growth factor 2	*igf2*	F: TCCTTTGTTTGTTGCCATTTG R: GAGTCCCATCCATTCTGTTG	NM_131433.1	59
Myostatin a	*mstna*	F: GGACTGGACTGCGATGAG R: GATGGGTGTGGGGATACTTC	AF019626.1	58
Thyroid hormone receptor alpha a	*thraa*	F: GGAAACAGAAGCGCAAGTTC R: TCTTCACAAGGCAGCTCTGA	NM_131396.1	52
BCL2 associated X apoptosis regulator a	*baxa*	F: CAACAAGATGGCATCACACC R: TGAACCCGCTCGTATATGAAA	NM_131562.2	60
Bcl-2 apoptosis regulator a	*bcl2a*	F: CCTTCAATAAAGCAGTGGAGGAA R: CGGGCTATCAGGCATTCAGA	NM_001030253.2	60
caspase 3, apoptosis-related cysteine peptidase a	*casp3a*	F: GTGCCAGTCAACAAACAAAG R: CATCTCCAACCGCTTAACG	NM_131877.3	60
Collagen type X alpha 1a	*col10a1a*	F: CCCATCCACATCACATCAAA R: GCGTGCATTTCTCAGAACAA	NM_001083827.1	60
Secreted phosphoprotein 1	*spp1*	F: GAGCCTACACAGACCACGCCAACAG R: GGTAGCCCAAACTGTCTCCCCG	NM_001002308.1	60
RUNX family transcription factor b	*runx2b*	F: GTGGCCACTTACCACAGAGC R: TCGGAGAGTCATCCAGCTT	NM_212862.2	60
Cytochrome P450, family 26 subfamily b polypeptide 1	*cyp26b1*	F: GCTGTCAACCAGAACATTCCC R: GGTTCTGATTGGAGTCGAGGC	NM_212666.1	60

## Data Availability

Data are contained within the article or Appendix A.

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
