# Peer review of "Mitigation of PFOA-Induced Developmental Toxicity in Danio rerio by Bacillus subtilis var. natto: Focus on Growth and Ossification"

_ijms, 2025, doi:10.3390/ijms26094261_

Round 1
Reviewer 1 Report
Comments and Suggestions for Authors
In the present manuscript, the authors investigated PFOA's developmental toxicity and the beneficial effects of probiotics on PFOA-induced developmental toxicity using zebrafish.
First of all, the objective of this experiment is ambiguous. The authors described a protective effect of probiotics on PFOA-exposed zebrafish, however, I don’t think the purpose of this paper was to improve the health of the zebrafish. The purpose and significance of this paper need to be clearly stated.
There is a lack of novelty. As described in Lines 48 and 80, the PFOA-induced growth and ossification defects and beneficial effect of probiotics Bacillus subtilis var. 80 natto on zebrafish larval growth and ossification had already been published. In this paper, the authors just showed that co-treatment of PFOA and probiotics partially restored the PFOA-induced growth and ossification defect. The authors need to highlight what is new in this paper.
In my opinion, this paper requires a lot of revision in content. The manuscript is not acceptable for publication in its present form.
Here, I summarized the major points to consider throughout the whole manuscript:
- Materials and Methods
4.1. Zebrafish husbandry and treatment (Line 256)
Describe how many larvae were used in total for each group.
Write the appropriate word in the place of [91] (Line 296) and delete “, 2007” (Line 304)
(Line 320) Please provide pictures of the agarose gel of RNA samples.
4.5. Gene expression analysis (Line 315)
How did you validate the specificity and efficiency of your primers? You need to run a relative standard curve to do this. Please follow MIQE guidelines: https://www.ncbi.nlm.nih.gov/pubmed/19246619.
Please describe in detail how the data were analyzed.
4.6. Statistical analysis
Why did you use “two-way ANOVA statistical analysis” instead of one-way ANOVA? (Line 332)
If you want to use two-way ANOVA, please explain in detail what were 2 factors and how the authors dealt with comparisons for the 2 factors. Please provide additional information such that the readers can understand your specific statistical comparisons.
Please rewrite the following sentence (Lines 332-335) to make it clearer: All the data were analyzed using two-way ANOVA statistical analysis followed by Tukey correction for multiple comparison, performed using the statistical software GraphPad Prism 8 Software (Inc., San Diego, CA, USA), with statistical significance set up at p < 0.05.
- Results
Write the full name of a.u. (Figure 1c) (Line 113)
There were many genes in real-time PCR data that did not differ between the control and treated groups.
I think there are other genes involved in zebrafish growth and development or ossification. Why didn’t you check them?
In your data, toxicity did not show a dose dependence. What do the authors think about this?
Author Response
REVIEWER 1
Open Review
( ) I would not like to sign my review report
(x) I would like to sign my review report
Quality of English Language
( ) The English could be improved to more clearly express the research.
(x) The English is fine and does not require any improvement.
Yes Can be improved Must be improved Not applicable
Does the introduction provide sufficient background and include all relevant references?
(x) ( ) ( ) ( )
Is the research design appropriate?
( ) (x) ( ) ( )
Are the methods adequately described?
( ) (x) ( ) ( )
Are the results clearly presented?
( ) ( ) (x) ( )
Are the conclusions supported by the results?
( ) ( ) (x) ( )
Comments and Suggestions for Authors
In the present manuscript, the authors investigated PFOA's developmental toxicity and the beneficial effects of probiotics on PFOA-induced developmental toxicity using zebrafish.
First of all, the objective of this experiment is ambiguous. The authors described a protective effect of probiotics on PFOA-exposed zebrafish, however, I don’t think the purpose of this paper was to improve the health of the zebrafish. The purpose and significance of this paper need to be clearly stated.
We would like to thank the reviewer for the comment. We agree with the Reviewer regarding the fact that the aim of the present study was not to improve zebrafish health by probiotic administration but investigate the mitigation capacity of this probiotic strain against the developmental, growth and ossification alterations induced by PFOA, from 3 days post fertilization (dpf) to 21 dpf. Regarding the significance of the present work, the research highlights how PFOA disrupts developmental processes by impairing growth, delaying ossification, and altering craniofacial cartilage formation and demonstrates the probiotic capacity to mitigate specific PFOA negative effects. To let clearly emerge the aim of the present study, the manuscript was revised in Line 97-99 in the Introduction and in Line 261-263 in the Discussion.
There is a lack of novelty. As described in Lines 48 and 80, the PFOA-induced growth and ossification defects and beneficial effect of probiotics Bacillus subtilis var. 80 natto on zebrafish larval growth and ossification had already been published. In this paper, the authors just showed that co-treatment of PFOA and probiotics partially restored the PFOA-induced growth and ossification defect. The authors need to highlight what is new in this paper.
We appreciate this valuable comment that will help us to improve the significance of our study. As stated by the reviewer, the effect of the selected probiotic strain on zebrafish ossification (10.3390/ijms23094748) has already been investigated but just up to 10 dpf, whereas in the present study we monitored development and ossification at 7, 14 and 21 dpf. This difference in the timing of probiotic treatment is crucial since we have encompassed different time points of ossification: early ossification with most skeletal elements being cartilaginous (7dpf), onset of mineralization when jaw bones and vertebrae start to show mineralization (14 dpf) and advanced ossification with most skeletal elements showing mineralization (21 dpf). Considering the toxicity assessment of PFOA exposure, a study by Wang et al., 2020 (10.1016/j.envint.2019.105317) investigated the toxicity of this pollutant (at the same concentrations used in our study) on zebrafish development until 5 dpf calculating the malformation rate (%) which comprise swim bladder inflation, yolk sac, and pericardial oedemas and spinal deformities. Therefore, as far as we are concerned, we are the first to study the impact of PFOA on zebrafish bone mineralization, specifically during three different times (7, 14 or 21 dpf). This study has been conducted at histological level (using alcian blue and alizarin red staining) and at molecular level, analyzing the expression of genes involved in ossification process. Still, we would like to highlight that the major novelty of this study is the assessment of the mitigating capacity of probiotics to counteract PFAS toxicity on bone mineralization, something that has not been studied before either in zebrafish or other model species.
Overall, this study not only advances knowledge on PFOA toxicity on ossification but also provides a promising foundation for developing intervention strategies to reduce pollutant-related developmental defects. We have now added this sentence at the end of the discussion to emphasize the novelty of our results (line 261-263).
In my opinion, this paper requires a lot of revision in content. The manuscript is not acceptable for publication in its present form.
Here, I summarized the major points to consider throughout the whole manuscript:
Materials and Methods
4.1. Zebrafish husbandry and treatment (Line 256)
Describe how many larvae were used in total for each group.
In this study a total of 3000 larvae were used. At 3 dpf, larvae were equally divided into 6 experimental groups in triplicate. This information has been added in the text of the manuscript in line 272 to line 275.
Write the appropriate word in the place of [91] (Line 296) and delete “, 2007” (Line 304)
Zarantoniello et al., was added before reference [91] in line 310-311 and “, 2007” was deleted in line 318. Both changes were highlighted in yellow in the text.
(Line 320) Please provide pictures of the agarose gel of RNA samples.
The Authors wish to thank the Reviewer for its comment. The Agarose gel run of all the RNA samples is provided in this rebuttal letter, see the Figure below. The RNA samples were run on 1% Agarose gel using Fast Red stain or Xpert Green stain.
4.5. Gene expression analysis (Line 315)
How did you validate the specificity and efficiency of your primers? You need to run a relative standard curve to do this. Please follow MIQE guidelines: https://www.ncbi.nlm.nih.gov/pubmed/19246619.
The efficiency of the primers is usually analyzed before their use by running a relative standard curve in RTq-PCR. The formula Efficiency = [10^(-1/Slope) - 1] x 100 is used to determine the efficiency of the primers as it is reported below as percentage (%) together with the slope and R2 values.
In addition, in M&M the correct igf2 primer sequences were provided in Table 1.
Please describe in detail how the data were analyzed.
RTq-PCR data were analyzed using the Gene Expression Analysis for iCycler iQ® Real-Time PCR Detection System provided by BioRad. This information has been added in line 343-344. The relative expression values calculated for each gene on the two housekeeping (rplp0 and rpl13a) was then analyzed using GraphPad Prism 8 Software (Inc., San Diego, CA, USA) as specified below regarding statistical analysis.
4.6. Statistical analysis
Why did you use “two-way ANOVA statistical analysis” instead of one-way ANOVA? (Line 332)
If you want to use two-way ANOVA, please explain in detail what were 2 factors and how the authors dealt with comparisons for the 2 factors. Please provide additional information such that the readers can understand your specific statistical comparisons.
Please rewrite the following sentence (Lines 332-335) to make it clearer: All the data were analyzed using two-way ANOVA statistical analysis followed by Tukey correction for multiple comparison, performed using the statistical software GraphPad Prism 8 Software (Inc., San Diego, CA, USA), with statistical significance set up at p < 0.05.
We thank the reviewer for pointing out the need for clarification regarding our statistical analysis. We chose to use two-way ANOVA because our experimental design involved two independent factors: the treatment (e.g., control, PFOA, probiotic, PFOA + probiotic) and the stage of development (e.g., 7-, 14-, and 21-days post-fertilization [dpf]). Although our main interest was to evaluate differences within each developmental stage (i.e., across treatments at the same time point), we used two-way ANOVA in GraphPad Prism 8.0 with the “compare columns (treatments), compare within rows (same developmental stage)” option. This allowed us to assess the main effect of treatment within each developmental stage, accounting for possible interaction effects between treatment and developmental stage. Although we did not directly compare across different stages, we applied appropriate post-hoc correction using Tukey’s test for multiple comparisons within each row. This approach ensured that the analysis respected the factorial design while focusing comparisons on treatment effects within each stage, which was the main biological question of interest in this study.
We have now revised the sentence in the manuscript for clarity in line 347-350 of Materials and Methods, as follows: "All data were analyzed using a two-way ANOVA with treatment and developmental stage as factors, followed by Tukey’s multiple comparisons test (GraphPad Prism 8.0, San Diego, CA, USA). Post-hoc comparisons were performed within each developmental stage to evaluate differences among treatments. Statistical significance was set at p < 0.05".
Results
Write the full name of a.u. (Figure 1c) (Line 113)
We have added this information to the figure caption of Figure 1 and Figure 4.
There were many genes in real-time PCR data that did not differ between the control and treated groups.
I think there are other genes involved in zebrafish growth and development or ossification. Why didn’t you check them?
Indeed, numerous genes are involved in the regulation of growth, development, and ossification in zebrafish, in this study, we focused on a set of key genes (thraa, col10a1a, runx2b, cyp26b1) selected based on their well-established and central roles in the specific biological processes including growth retardation and skeletal malformations. These genes have been consistently reported in the literature to respond to endocrine-disrupting compounds and to mediate pathways involving thyroid hormone regulation, ossification, chondrogenesis, and retinoic acid signaling [https://doi.org/10.3389/fendo.2019.00156 , https://www.mdpi.com/2218-273X/14/2/139; https://pubmed.ncbi.nlm.nih.gov/18927157/].
We recognize that some of these genes did not show statistically significant changes in expression under our experimental conditions, however, their inclusion was important to confirm that the molecular effects we observed were specific, and not a result of widespread, non-targeted gene expression disruption. The absence of change in some transcripts may reflect either the resilience of certain regulatory pathways, or differences in temporal expression windows that were not captured at the selected time points. Our aim was to provide an initial mechanistic insight into how PFOA and probiotics may influence development. We agree that a broader transcriptomic or pathway-focused analysis (e.g., via RNA-seq or microarrays) could further expand our understanding providing information regarding other genes potentially involved in the process and we would consider it in future studies.
In your data, toxicity did not show a dose dependence. What do the authors think about this?
We appreciate the reviewer’s insightful observation. Indeed, our data suggests a lack of traditional monotonic dose dependence in some endpoints. This observation is consistent with emerging evidence on non-monotonic dose-response (NMDR) relationships, particularly in the context of endocrine-disrupting chemicals (EDCs).
EDCs are known to exert biological effects at low doses through mechanisms such as receptor binding, feedback disruption, and differential pathway activation, which may not follow a linear or monotonic pattern. Non-monotonic responses, including U-shaped or inverted U-shaped curves, have been documented for various EDCs such as bisphenol A and phthalates (10.2203/dose-response.13-020.Vandenberg ; 10.1007/s00204-022-03409-9). These effects are thought to arise from complex hormonal feedback systems, receptor saturation, and changes in receptor expression at different concentrations.
Given these dynamics, the absence of dose dependence in our results does not necessarily reflect experimental error or lack of effect but rather aligns with the growing body of literature on NMDR compounds in endocrine biology. We have added a sentence of this point in the revised manuscript (see lines 244-246 of the discussion and line 356 of conclusion) to provide context and highlight its relevance to interpreting our findings.
Submission Date
13 March 2025
Date of this review
31 Mar 2025 03:14:21

Reviewer 2 Report
Comments and Suggestions for Authors
General Comments
There are serious concerns about the methodology used in this study, including:
Evaluating and comparing probiotic administration in zebrafish embryos exposed to PFOA is problematic because the cause(s) of PFOA-induced larval growth retardation and ossification defect are not clear. In addition, there was no confirmation that the embryos in this study had not been exposed to PFOA prior to the study. Thus, any co-morbid conditions that the embryos in this study may have are confounders that would be expected to affect the results of these exposure tests.
It is also unclear whether the embryos in this study were all exposed to the same PFASs prior to the study. This is important because many adverse effects are expected to vary depending on the level of exposure. Therefore, PFASs exposure is another potential confounding variable in this study.
Specific Comments
Line45-47
Negative effects on humans or animals?
Line48-49
If there could be more detail for exposure levels, that would be great.
Line 50-51
Again, more detailed information on exposure levels for endocrine disrupting activity is needed.
Line 260-267
No information on the sample size in each group? Do you consider the effect size?
Author Response
REVIEWER 2
Open Review
(x) I would not like to sign my review report
( ) I would like to sign my review report
Quality of English Language
( ) The English could be improved to more clearly express the research.
(x) The English is fine and does not require any improvement.
Yes Can be improved Must be improved Not applicable
Does the introduction provide sufficient background and include all relevant references?
( ) ( ) (x) ( )
Is the research design appropriate?
( ) ( ) (x) ( )
Are the methods adequately described?
( ) ( ) (x) ( )
Are the results clearly presented?
( ) ( ) (x) ( )
Are the conclusions supported by the results?
( ) ( ) (x) ( )
Comments and Suggestions for Authors
General Comments
There are serious concerns about the methodology used in this study, including:
Evaluating and comparing probiotic administration in zebrafish embryos exposed to PFOA is problematic because the cause(s) of PFOA-induced larval growth retardation and ossification defect are not clear. In addition, there was no confirmation that the embryos in this study had not been exposed to PFOA prior to the study. Thus, any co-morbid conditions that the embryos in this study may have are confounders that would be expected to affect the results of these exposure tests.
We appreciate the reviewer’s comment and the opportunity to clarify these important points. First, regarding the mechanisms underlying PFOA-induced growth retardation and ossification defects: while it is true that these pathways are complex and not fully understood, our study contributes novel evidence that helps illuminate these effects through both morphological and molecular endpoints. Specifically, we demonstrated that exposure to PFOA resulted in altered larval growth (as evidenced by reduced standard length and wet weight), skeletal abnormalities, and jaw malformations. Importantly, our molecular analyses revealed dysregulation of genes involved in growth and ossification (e.g., thraa, col10a1a, runx2b, and cyp26b1), providing mechanistic insights that align with phenotypic observations. S imilarly, deregulation of runx2b and col10a1a, master regulators of bone formation, further supports the notion that PFOA impairs ossification through disruption of cartilage-to-bone transition pathways.
Regarding the concern about potential pre-exposure to PFOA, we would like to clarify that all embryos used in this study were obtained from a well-established laboratory breeding colony housed in a zebrafish facility that operates with a reverse osmosis water system. While we do not perform routine chemical testing of the water for PFAS contamination, the use of reverse osmosis water significantly reduces the likelihood of background contamination by such persistent pollutants. This information has been added to the Materials and Methods section (lines 272-275 of the revised manuscript for clarity.
Finally, while we acknowledge that unknown co-morbid conditions can theoretically confound developmental studies, all embryos used were obtained from synchronized breedings under controlled conditions and handled identically across groups. The consistency of the results across multiple biological endpoints and gene expression patterns strengthens the reliability of our findings and supports the conclusion that probiotic administration can mitigate specific PFOA-induced developmental effects. We have revised the Materials and Methods section (lines 272-275) accordingly to address these points more clearly.
It is also unclear whether the embryos in this study were all exposed to the same PFASs prior to the study. This is important because many adverse effects are expected to vary depending on the level of exposure. Therefore, PFASs exposure is another potential confounding variable in this study.
We appreciate the reviewer’s thoughtful comment and agree that controlling for unintentional PFAS exposure is critical for studies assessing the developmental effects of environmental contaminants like PFOA. To address this concern, we clarify that all zebrafish embryos used in this study were obtained from a controlled laboratory breeding colony maintained in Tecniplast aquatic housing systems, which are manufactured using LEXAN™ resin 104H-11204. According to the manufacturer's documentation, this material is compliant with stringent EU and FDA food-contact regulations, including Commission Regulation (EU) No. 10/2011 and FDA 21 CFR 177.1580. While the resin contains some regulated substances such as 2,2-bis(4-hydroxyphenyl)propane (a BPA-related compound), these components have defined specific migration limits that are well below toxicological thresholds and the material is rated as suitable for food contact at temperatures up to 100 °C.
Moreover, the aquatic system is supplied with reverse osmosis water, which significantly reduces the risk of contamination by PFAS and other environmental pollutants. Although routine testing for PFAS in our facility water was not performed, the use of reverse osmosis water and certified food-grade materials for housing substantially minimizes the likelihood of background PFAS exposure. All embryos were bred, maintained, and treated under the same standardized conditions, ensuring consistency across all experimental groups.
To further improve transparency, we have added a statement in the Materials and Methods section (line 272-275 and line 282-283) describing the use of reverse osmosis water, and the uniform rearing conditions, which help mitigate the potential for unintentional PFAS exposure as a confounding variable.
Specific Comments
Line45-47
Negative effects on humans or animals?
The references reported in lines 45 to 47 all refer to zebrafish (Danio rerio). This information has been added to the text in line 47.
Line48-49
If there could be more detail for exposure levels, that would be great.
This information has been added to the text in lines 48 to 50.
Line 50-51
Again, more detailed information on exposure levels for endocrine disrupting activity is needed.
This information has been added to the text in lines 50 to 56.
Line 260-267
No information on the sample size in each group? Do you consider the effect size?
In this study a total of 3000 larvae were used. At 3 dpf, larvae were equally divided into 6 experimental groups in triplicate. This information has been added to the text at line 264 and line 266. In addition, the effect size has been considered. Given our sample size of 3000 zebrafish embryos, with 500 per group and multiple time points (7, 14, and 21 dpf), effect size is a critical factor in assessing biological relevance. Depending on the statistical comparisons made (e.g., between treatment and control groups or across time points), we have evaluated effect sizes using appropriate metrics such as Cohen’s d for pairwise comparisons, eta squared (η2) for ANOVA, or odds ratios for categorical outcomes. Our large sample size allows us to detect even small effect sizes, ensuring that our findings are not only statistically significant but also biologically meaningful.
Submission Date
13 March 2025
Date of this review
24 Mar 2025 09:50:48

Round 2
Reviewer 1 Report
Comments and Suggestions for Authors
- I recommend adding the rationale for the sampling timing specifically to the main text, as you described.
"There is a lack of novelty. As described in Lines 48 and 80, the PFOA-induced growth and ossification defects and beneficial effect of probiotics Bacillus subtilis var. 80 natto on zebrafish larval growth and ossification had already been published. In this paper, the authors just showed that co-treatment of PFOA and probiotics partially restored the PFOA-induced growth and ossification defect. The authors need to highlight what is new in this paper.
We appreciate this valuable comment that will help us to improve the significance of our study. As stated by the reviewer, the effect of the selected probiotic strain on zebrafish ossification (10.3390/ijms23094748) has already been investigated but just up to 10 dpf, whereas in the present study we monitored development and ossification at 7, 14 and 21 dpf. This difference in the timing of probiotic treatment is crucial since we have encompassed different time points of ossification: early ossification with most skeletal elements being cartilaginous (7dpf), onset of mineralization when jaw bones and vertebrae start to show mineralization (14 dpf) and advanced ossification with most skeletal elements showing mineralization (21 dpf). Considering the toxicity assessment of PFOA exposure, a study by Wang et al., 2020 (10.1016/j.envint.2019.105317) investigated the toxicity of this pollutant (at the same concentrations used in our study) on zebrafish development until 5 dpf calculating the malformation rate (%) which comprise swim bladder inflation, yolk sac, and pericardial oedemas and spinal deformities. Therefore, as far as we are concerned, we are the first to study the impact of PFOA on zebrafish bone mineralization, specifically during three different times (7, 14 or 21 dpf). This study has been conducted at histological level (using alcian blue and alizarin red staining) and at molecular level, analyzing the expression of genes involved in ossification process. Still, we would like to highlight that the major novelty of this study is the assessment of the mitigating capacity of probiotics to counteract PFAS toxicity on bone mineralization, something that has not been studied before either in zebrafish or other model species.
Overall, this study not only advances knowledge on PFOA toxicity on ossification but also provides a promising foundation for developing intervention strategies to reduce pollutant-related developmental defects. We have now added this sentence at the end of the discussion to emphasize the novelty of our results (line 261-263).
2. How about providing pictures of the agarose gel of RNA samples in supplementary figure?
3. Please explain any abbreviations that appear at the beginning with the full name (Line 147-148).
Author Response
Comments and Suggestions for Authors
I recommend adding the rationale for the sampling timing specifically to the main text, as you described.
"There is a lack of novelty. As described in Lines 48 and 80, the PFOA-induced growth and ossification defects and beneficial effect of probiotics Bacillus subtilis var. 80 natto on zebrafish larval growth and ossification had already been published. In this paper, the authors just showed that co-treatment of PFOA and probiotics partially restored the PFOA-induced growth and ossification defect. The authors need to highlight what is new in this paper.
We appreciate this valuable comment that will help us to improve the significance of our study. As stated by the reviewer, the effect of the selected probiotic strain on zebrafish ossification (10.3390/ijms23094748) has already been investigated but just up to 10 dpf, whereas in the present study we monitored development and ossification at 7, 14 and 21 dpf. This difference in the timing of probiotic treatment is crucial since we have encompassed different time points of ossification: early ossification with most skeletal elements being cartilaginous (7dpf), onset of mineralization when jaw bones and vertebrae start to show mineralization (14 dpf) and advanced ossification with most skeletal elements showing mineralization (21 dpf). Considering the toxicity assessment of PFOA exposure, a study by Wang et al., 2020 (10.1016/j.envint.2019.105317) investigated the toxicity of this pollutant (at the same concentrations used in our study) on zebrafish development until 5 dpf calculating the malformation rate (%) which comprise swim bladder inflation, yolk sac, and pericardial oedemas and spinal deformities. Therefore, as far as we are concerned, we are the first to study the impact of PFOA on zebrafish bone mineralization, specifically during three different times (7, 14 or 21 dpf). This study has been conducted at histological level (using alcian blue and alizarin red staining) and at molecular level, analyzing the expression of genes involved in ossification process. Still, we would like to highlight that the major novelty of this study is the assessment of the mitigating capacity of probiotics to counteract PFAS toxicity on bone mineralization, something that has not been studied before either in zebrafish or other model species.
Overall, this study not only advances knowledge on PFOA toxicity on ossification but also provides a promising foundation for developing intervention strategies to reduce pollutant-related developmental defects. We have now added this sentence at the end of the discussion to emphasize the novelty of our results (line 261-263).
We thank the reviewer for the valuable comments. We have modified the text in lines 99-103 in this way “To test this hypothesis, integrated morphological and molecular studies were conducted over different crucial developmental stages, encompassing different time points of ossification: early ossification with most skeletal elements being cartilaginous (7dpf), onset of mineralization when jaw bones and vertebrae start to show mineralization (14 dpf) and advanced ossification with most skeletal elements showing mineralization (21 dpf)” highlighting in red the changes in the text.
- How about providing pictures of the agarose gel of RNA samples in supplementary figure?
Supplementary Figure 4 was added to provide the agarose gel run of RNA samples in the supplementary materials section and was cited in the line 338 of the manuscript.
- Please explain any abbreviations that appear at the beginning with the full name (Line 147-148).
All the abbreviations regarding groups names were added in the text between lines 130 to 152 and highlighted in red.
Reviewer 2 Report
Comments and Suggestions for Authors
I appreciate the author's reply. However, I would like to ask whether each group's embryo mortality rates are similar (Line 274).
Author Response
Comments and Suggestions for Authors
I appreciate the author's reply. However, I would like to ask whether each group's embryo mortality rates are similar (Line 274).
We thank the reviewer for this valuable comment. While embryo mortality was not explicitly monitored throughout the experiment, we collected samples at 7, 14, and 21 days post-fertilization, and consistently observed similar numbers of viable embryos across all treatment groups. At the end of the experiment, we successfully obtained samples from each group, suggesting there were no notable differences in mortality. We agree that tracking mortality more systematically is important and will address this more thoroughly in future studies